# Potential Involvement of *ewsr1-w* Gene in Ovarian Development of Chinese Tongue Sole, *Cynoglossus semilaevis*

**DOI:** 10.3390/ani12192503

**Published:** 2022-09-20

**Authors:** Peng Cheng, Zhangfan Chen, Wenteng Xu, Na Wang, Qian Yang, Rui Shi, Xihong Li, Zhongkai Cui, Jiayu Cheng, Songlin Chen

**Affiliations:** 1Laboratory for Marine Fisheries Science and Food Production Processes, Qingdao National Laboratory for Marine Science and Technology, Yellow Sea Fisheries Research Institute, Chinese Academy of Fishery Sciences (CAFS), Qingdao 266071, China; 2School of Fisheries and Life Science, Shanghai Ocean University, Shanghai 201306, China; 3Engineering and Technology Center for Flatfish Aquaculture of Tangshan, Tangshan Weizhuo Aquaculture Co., Ltd., Tangshan 063202, China

**Keywords:** *Cynoglossus semilaevis*, *ewsr1-w*, ovarian development, Mafk, RNA interference

## Abstract

**Simple Summary:**

Sexual dimorphism is a phenomenon commonly existing in animals. Chinese tongue sole *Cynoglossus semilaevis* is an economical marine fish with obvious female-biased size dimorphism. So, it is important to explore the molecular mechanism beyond gonadal development for sex control in aquaculture industry. RNA-binding protein Ewing Sarcoma protein-like (*ewsr1*) gene is important for mouse gonadal development and reproduction, however there are limited studies on this gene in teleost. In this study, two *ewsr1* genes were cloned and characterized from *C. semilaevis*. The *ewsr1-w gene*, located in W chromosomes, showed female-biased expression during *C. semilaevis* gonadal development. In addition, knock-down effect and transcriptional regulation of *Cs-ewsr1-w* further suggested its essential role in ovarian development. This study broadened our understanding on *ewsr1* function in teleost, and provided genetic resources for the further development of sex control breeding techniques in *C. semilaevis* aquaculture.

**Abstract:**

Ewsr1 encodes a protein that acts as a multifunctional molecule in a variety of cellular processes. The full-length of *Cs-ewsr1-w* and *Cs-ewsr1-z* were cloned in Chinese tongue sole (*Cynoglossus semilaevis*). The open reading frame (ORF) of *Cs-ewsr1-w* was 1,767 bp that encoded 589 amino acids, while *Cs-ewsr1-z* was 1,794 bp that encoded 598 amino acids. Real-time PCR assays showed that *Cs-ewsr1-w* exhibited significant female-biased expression and could be hardly detected in male. It has the most abundant expression in ovaries among eight healthy tissues. Its expression in ovary increased gradually from 90 d to 3 y with *C. semilaevis* ovarian development and reached the peak at 3 y. After *Cs-ewsr1-w* knockdown with siRNA interference, several genes related to gonadal development including *foxl2*, *sox9b* and *pou5f1* were down-regulated in ovarian cell line, suggesting the possible participation of *Cs-ewsr1-w* in *C. semilaevis* ovarian development. The dual-luciferase reporter assay revealed that the -733/-154 bp *Cs-ewsr1-w* promoter fragment exhibited strong transcription activity human embryonic kidney (HEK) 293T cell line. The mutation of a MAF BZIP Transcription Factor K (Mafk) binding site located in this fragment suggested that transcription factor Mafk might play an important role in *Cs-ewsr1-w* basal transcription. Our results will provide clues on the gene expression level, transcriptional regulation and knock-down effect of *ewsr1* gene during ovarian development in teleost.

## 1. Introduction

Sexual dimorphism is widespread in mammals, fish, birds and reptiles that characterized by body size, physiological and color differences between females and males [1,2,3,4,5,6]. This phenomenon has been found in a lot of fish species, of which Chinese tongue sole *Cynoglossus semilaevis* shows typically female-biased size dimorphism [2]. Female *C. semilaevis* can reach over twice in size and weight of males at the same age [7].

Sexual dimorphism is mainly resulted by genetic selection during the evolutionary process and is the consequence of differential expression of sex-biased genes in development stages [8,9,10]. The previous transcriptome analysis revealed thousands of sex-biased genes in somatotropic and reproductive tissues of *C. semilaevis* [11]. *Cyp19a* gene was expressed higher in ovary than testis and rose along the gonadal development, implying its participation in sex determination [12]. *Dhcr24* (24-dehydrocholesterol reductase) gene, involved in steroid hormones and PI3K/Akt pathway and IGF-1 system, had the highest expression in liver and gonad of females [13]. The female-biased gonadal gene *igfbp7* (insulin-like growth factor binding protein 7) might be involved in growth regulation of *C. semilaevis* by influencing insulin-like growth factor 1 receptor (*igf1r*), serine/threonine kinase 1 (*akt*) and NFκB (the nuclear factor kappa B) signal [14]. To better understand the molecular mechanism refining sex determination and differentiation in *C. semilaevis*, comparative transcriptome analysis was performed to reveal 156 genes correlated with ovary differentiation, including RNA-binding protein Ewing Sarcoma protein-like (*ewsr1*) gene [15].

The *ewsr1* gene encodes a multifunctional RNA binding protein that regulates transcription and RNA splicing by interacting with other proteins and other cellular processes [16,17,18]. It is one of Translocated in liposarcoma, Ewing’s sarcoma and TATA-binding protein-associated factor 15 (TET, also named as FET) protein family members that also contains Fused in Sarcoma (FUS) and TATA-box binding protein Associated Factor 15 (TAF15) [17]. These proteins share high homology amino acid sequences in vertebrates [19]. EWSR1 regulates gene transcription by interacting with CREB binding proteins, basic transcription factors (TFs) TFIID and RNA polymerase II [17,20]. In zebrafish, *ewsr1a* and *ewsr1b* were required for mitotic stability and cellular survival in central nervous system (CNS) during early embryonic development [21]. Moreover, *ewsr1* regulated the transcription of *HNF4*, *oct4* and *BRN3A*, which involved with development and hormone regulation [16,22,23,24,25,26,27]. The offspring of *ewsr1*-deficient mice caused the abnormal gonadal development and subsequently sterile [22,28]. However, there has been rare focus on its function in gonadal development in teleost.

In *C. semilaevis* genome, it was found that two allele genes of *ewsr1* were located on chromosomes W and Z, named as *Cs-ewsr1-w* and *Cs-ewsr1-z*, respectively. Based on the transcriptome dataset, we cloned and characterized two genes. *Cs-ewsr1-w* gene was chosen for further analysis on its transcriptional regulation and knock-down effect. These results could improve our understanding on the role of *ewsr1* genes in *C. semilaevis.*

## 2. Materials and Methods

### 2.1. Ethics Approval

All the animal experiments were performed under the inspection of Yellow Sea Fisheries Research Institute’s animal care and use committee (Approval number, YSFRI-2022023). MS222 (Sigma-Aldrich, Oakville, ON, Canada) was used for anesthesia to minimize fish suffering (solubilized in seawater, final concentration 20 mg/L, fish was treated for 5 min) during experimental procedure [29]. The 293T cell line was purchased from ATCC (CRL-3216^TM^) (American Type Culture Collection, Manassas, VA, USA). The *C. semilaevis* ovarian cell line was previously established and cultured in our laboratory [30].

### 2.2. Samples Collection

All fish samples used in this experiment have been approved by the Care and Use of Laboratory Animals of the Chinese Academy of Fishery Sciences. Before sampling, genomic DNA was extracted from cut fins by TIANamp Marine Animals DNA Kit (TIANGEN, Beijing, China) for genetic sex identification by using PCR amplification with primers sex-F and sex-R (Table 1) [31]. After anesthesia with MS-222 (20 mg/L) [29], gonads were dissected from different developmental stages of *C. semilaevis*, including 90-day post hatching (90 d), 6-month post hatching (6 m), and 1.5-year post hatching (1.5 y). Six females and males were sampled at each stage. Brain, gonad, liver, spleen, heart, kidney, intestine and muscle were collected from three 3 y females and males. Tissues were put into RNAwait RNAlater solution (Solarbio, Beijing, China) quickly and stored in a refrigerator at −80 °C for subsequent RNA extraction.

### 2.3. Gene Cloning of Cs-ewsr1-w and Cs-ewsr1-z

RNA was extracted from each sample by using Trizol Reagent (Invitrogen, Carlsbad, CA, USA). The quality and quantity of RNA was checked with agarose gel electrophoresis and P100 Series Spectrophotometers (Pultton, San Jose, CA, USA). The first strand cDNA was synthesized with 800 ng RNA as the template by using PrimeScript RT Kit with gDNA eraser (TaKaRa, Tokyo, Japan). Gene specific primers for gene cloning were designed by Primer Premier 5.0 (Table 1) based on partial sequences of *ewsr1-w* and *ewsr1-z* from *C. semilaevis* genome [32]. The mixed cDNA of females and males was used as the template for PCR amplification. The 25 μL PCR mixture contained 12.5 μL Ex *Taq* Mix (TaKaRa, Tokyo, Japan), 0.5 μL forward/ reverse primers, and 1 μL cDNA template. The PCR program was set as follows: 95 °C for 5 min, 40 cycles of 95 °C for 30 s, 55 °C for 30 s, and 72 °C for 30 s, and 72 °C for 10 min. PCR products were purified by FastPure Gel DNA Extraction Mini Kit (Vazyme, Nanjing, China), connected to pEASY-T1 vector, transformed, and sequenced in Beijing Ribioco Biotechnology Co., Ltd. (Ribioco, Beijing, China). The 3′ and 5′ untranslated regions (UTR) were amplified by using SMARTer RACE 5′/3′ Kit (TaKaRa, Tokyo, Japan) with the primers listed in Table 1 and the PCR program was followed as mentioned above.

### 2.4. Characterization of Cs-ewsr1-w and Cs-ewsr1-z

The characters including open reading frame (ORF), amino acid sequence, molecular weight, protein domains, and phosphorylation sites were predicted and analyzed by DNAstar (V7.1.0) (Bioinformatics Software, Madison, WI, USA), SMART (V9.0, Letunic et al. [33], Heidelberg, Germany) (http://smart.embl.de/, accessed on 26 March 2022), and NetPhos-3.1 (https://services.healthtech.dtu.dk/service.php?NetPhos-3.1) (Department of Health Technology, Technical University of Denmark, Kongens Lyngby, Denmark). The BLASTP Program (https://blast.ncbi.nlm.nih.gov/Blast.cgi?PROGRAM=blastp&PAGE_TYPE=BlastSearch&LINK_LOC=blasthome) (National Center for Biotechnology Information, National Institutes of Health, Bethesda, MD, USA) was used for multiple sequence alignment. The MEGA X (Kumar et al. [34], Philadelphia, PA, USA) was used to construct phylogenetic tree by neighbour-joining algorithm (NJ). The NCBI accession numbers of amino acid sequences used in this study were listed in Table 2.

### 2.5. Gene Expression Patterns of Cs-ewsr1-w and Cs-ewsr1-z in Different Tissues and Stages

The expressions of *Cs-ewsr1-w* and *Cs-ewsr1-z* in different development stages and tissues were analyzed with gene specific primers (Table 1) via qPCR assays on a 7500 Fast Real Time PCR platform (Applied Biosystems, Foster City, CA, USA). *β*-actin was set as the internal control. The 20 μL reactions contained 10 μL SYBR Premix Ex TaqTM (TaKaRa, Tokyo, Japan), 2 μL cDNA, 0.4 μL of each sense and anti-sense primers, and 0.4 μL ROX Dye II. The qPCR program was set as default settings followed by the dissociation curve, that is, 95 °C for 30 s, 40 cycles of 95 °C for 5 s, 60 °C for 30 s. The relative mRNA expression of *Cs-ewsr1-w* and *Cs-ewsr1-z* were processed by using 2^−ΔΔCt^ method [35]. The data were analyzed by one-way ANOVA followed by Duncan’s multiple comparison in SPSS 25.0 (IBM Corp., Armonk, NY, USA), and the differences were considered significant when *p* < 0.05.

### 2.6. Promoter Activities Analysis of Cs-ewsr1-w

Based on the TFs prediction, six promoter plasmids with luciferase report were constructed by serial-deletion to detect the promoting activity of regulatory elements of *Cs-ewsr1-w*. The primers were listed in Table 1. The fragments were inserted into pGL3-basic vector (Promega, Madison, WI, USA) for the recombinant plasmid construction of pGL3-*Cs-ewsr1-w-*F1~F6 by using TSV-S1 Trelief^®^ SoSoo Cloning Kit (Tsingke, Beijing, China).

Human embryonal kidney (HEK) 293T cells were maintained in DME/F-12 containing 10% fetal bovine serum (FBS, Gibco, New York, NY, USA) and 1% bFGF (Invitrogen, Carlsbad, CA, USA) in 5% CO_2_ at 37 °C. The pGL3-*Cs-ewsr1-w*~F1~F6 were transfected into HEK293T cells by using Lipo8000TM Transfection Reagent (Beyotime, Shanghai, China). Meanwhile, pGL3-basic and PGL3-control plasmids were used as the negative control and the positive control, respectively. The pRL-TK plasmid was transfected at the mean time as the internal reference. Dual Luciferase Reporter Gene Assay Kit (Beyotime, Shanghai, China) was employed to measure the promoter activities. Each experiment was performed in triplicates following the standard protocol provided by the manufacturer. Data obtained from Varioskan Flash spectral scanning multimode reader (Thermo Fisher Scientific, Vantaa, Finland) were analyzed by LSD (Least-significant difference) in SPSS 25.0, and the significance was regarded at *p* < 0.05.

The TF binding sites were predicted by PROMO (http://alggen.lsi.upc.es/, accessed on 2 April 2022) (Messeguer et al. [36], Barcelona, Spain) and JASPAR 2022 (https://jaspar.genereg.net/, accessed on 2 April 2022) (Castro-Mondragon et al. [37], Oslo, Norway). Nucleotides were mutated within TF binding sites (Mafk, c-MYC, MAC1, and POU1F1a-binding sites) following the protocols of Fast Site-Directed Mutagenesis Kit (TIANGEN, Beijing, China). After successful mutations were confirmed by sequencing, transfection and dual luciferase assays detection were performed as mentioned above.

### 2.7. The Knockdown Effect of Cs-ewsr1-w siRNA in C. semilaevis Ovarian Cells

The specific siRNA of *Cs-ewsr1-w* gene, the negative control siRNA, and siR transfect control (5cy3) were synthesized in Sangon Biotech (Sangon, Shanghai, China). *C. semilaevis* ovarian cells were cultured in L-15 medium supplemented with 1% bFGF and 15% FBS at 24 °C. *Cs-ewsr1-w* siRNA was transfected into the cells by using riboFECT^TM^ CP Transfection Kit (Ribobio, Beijing, China) following the protocol described in the previous study [13]. Three replicates were set for both *Cs-ewsr1-w-*siRNA and negative control (NC) groups. At 48 h post transfection, the cellular status and the florescence of 5cy3-transfected cells would be checked. When it reached 90–95% of cell confluency and the percentage of transfection reached ~80%, it would be a good timing to harvest ovarian cells for the following experiments. After cell collection and RNA extraction, reverse transcription and qPCR assays were performed following the methods mentioned above. The relative expression levels of sex-related genes, such as Forkhead Box L2 (*foxl2*), SRY-box transcription factor 9b (*sox9b*), POU Class 5 Homeobox 1 (*pou5f1*) were measured with the primers listed in Table 1.

## 3. Results

### 3.1. Gene Cloning and Characterization of Cs-ewsr1s

*Cs-ewsr1-w* (GenBank accession no. 103397238) located in W chromosomes, with the full length of 2297 bp containing the ORF region of 1767 bp encoding 588 amino acids (Figure 1A). Functional domain prediction showed that Cs-Ewsr1-W contained RNA recognition motif in 313–393 residues and a Ran binding protein zinc finger domain in 454–480 residues. *Cs-ewsr1-z* (GenBank accession no. 103398620), located in Z chromosome, was 2230 bp in full length. It contains 1794 bp ORF region encoding 598 amino acids (Figure 1B). The same functional domains were located at 313–399 and 460–486 in Cs-Ewsr1-Z, respectively.

The phylogenetic tree was constructed by using Ewsr1 proteins from 17 different species. The results showed that two Cs-Ewsr1 proteins were clustered together and then embedded with the teleost clade with *Paralichthys olivaceus*, *Scophthalmus maximus* and *Poecilia formosa*. The mammalian and the other species were clustered together (Figure 2). RNA recognition motif and Ran binding protein zinc finger domain were conserved in all the aligned species, including teleost, amphibians, and mammalians.

### 3.2. The Expression Patterns of Cs-ewsr1s in Different Tissues and Developmental Stages

qPCR assays revealed that the sex-biased expression patterns of two *Cs-ewsr1*s in Chinese tongue sole. *Cs-ewsr1-w* was expressed in all tissues of female tongue sole with the highest expression in gonad, but was hardly detected in any tissue of male tongue sole (Figure 3A). In comparison, *Cs-ewsr1-z* was prevalently expressed in all tissues of female and male tongue sole (Figure 3B). It exhibited the highest expression in male gonad, followed by female gonad, brains of male and female, livers of male and female, and female heart and intestine (Figure 3).

*Cs-ewsr1-w* gradually increased with ovary development, and reached the peak at 3 y (Figure 4A). However, its expression was hardly detected during testis development because of no expression in testis. *Cs-ewsr1-z* gene was expressed in all tested developmental stages of ovaries and testes. Its expression was relatively low in 6 m female and male, and was significantly higher in testes of 1.5 y male (Figure 4B).

### 3.3. Promoter Activity of Cs-ewsr1-w Detection and Analysis

The *Cs-ewsr1-w* promoter sequence of 2690 bp (−2590/+99) was cloned by genomic DNA with specific primers *Cs-ewsr1-w*-P-F/R (Table 1). *Cs-ewsr1-w* promoter region had only one CpG island, which was located from −1215 to −1101 bp.

A series of promoter fragments with different length deletion were generated to explore the promoter activity of *Cs-ewsr1-w* gene. The promoter activities of all *Cs-ewsr1-w* fragments were significantly higher than that of pGL3-basic (*p* < 0.05, Figure 5), among which the activity of *Cs-ewsr1-w*-P-F2/R fragment was the highest. The relative activity significantly decreased by 2.7-fold from *Cs-ewsr1-w*-P-F5/R fragment to *Cs-ewsr1-w*-P-F6/R fragment (*p* < 0.05, Figure 5), indicated region −733 to −154 positively affected the promoter activity of *Cs-ewsr1-w* gene. Similarly, the positive effect was detected from other two region as well, which were −2190 to −1692 bp and −154 to +99 bp (Figure 5).

Prediction of TF binding sites in the -733/-154 bp interval of *Cs-ewsr1-w* promoter revealed numerous TFs binding sites, including STAT4, HOXA3, c-Myc, GAGA factor, MAC1, POU1F1a, Mafk, and PRA (Figure 6). The interval containing TF Mafk, c-MYC, MAC1 and POU1F1a were mutated and transfected into 293T cells for detection after 48 h. The results indicated mutation of Mafk binding site led to the significant decrease by 46% in the activity of *Cs-ewsr1-w*-P-F5/R fragment (*p* < 0.05, Figure 7), which showed no significant difference compared with the activity of *Cs-ewsr1-w*-P-F6/R fragment. Mutations on other TF binding sites showed no significant effect (Figure 7).

### 3.4. Expression Patterns of Sex-Related Genes in Cs-ewsr1-w Knockdown Ovarian Cells

*Cs-ewsr1-w* expression was significantly reduced by 80% (*p* < 0.05) after in vitro *Cs-ewsr1-w* siRNA interference (RNAi), while no significant variation of *Cs-ewsr1-z* gene was detected. The down-regulation of sex-related genes was detected after *Cs-ewsr1-w* RNAi (*p* < 0.05), including *foxl2*, *sox9b* and *pou5f1*. Among that, *foxl2* expression significantly dropped by18.8-fold compared with the control group. The expression levels of *sox9b* and *pou5f1* were significantly reduced by half (*p* < 0.05, Figure 8).

## 4. Discussion

Ewsr protein, a key player in cancer, is involved in RNA metabolism and DNA repair [38]. However, studies on teleost EWS protein are rare. Based on the comparative transcriptome analysis on early developmental stages of gonad in *C. semilaevis*, *ewsr1-w* was specifically expressed in ovary and continuously up-regulated with ovarian differentiation [15]. The amino acid sequences of Cs-Ewsr1-w and Cs-Ewsr1-z proteins have high similarity of 91.41%, both of which contained the conserved RNA recognition motif and Ran binding protein-zinc finger domain, suggesting that Cs-Ewsr1s might have similar function with Ewsr1s in other vertebrates. RNA recognition motif is associated with the interaction of protein with RNA. Meanwhile, this protein has a variable number of RGG (arginine-glycine-glycine) repeats that are regarded as a RNA-binding region as well [39]. Based on the phylogenetic analysis, the sequences we obtained from *C. semilaevis* fell in a well-supported clade, suggesting that we obtained the *ewsr1* gene orthologs.

Based on our qPCR results, *Cs-ewsr1-w* gene was uniquely expressed in females with the highest transcriptional level in ovary. Its expression increased gradually with ovarian development from 90 d to 3 y. These results indicated the possible involvement of *Cs-ewsr1-w* gene in ovarian development. After *Cs-ewsr1-w* gene expression was interfered in the ovarian cells of *C. semilaevis*, several gonadal development-related genes were down-regulated, including *foxl2*, *sox9b* and *pou5f1*. *Foxl2* gene, belonging to winged helix transcription factor, is one of the crucial players in ovarian development [40]. Many studies have shown that this gene functions in sex differentiation and gonadal development in teleost [41,42,43,44]. *C. semilaevis foxl2* was significantly expressed in 12-month old ovary of phase II fish ovary development stages, suggesting its possible involvement in oocyte development [45]. *Sox9b*, another important gene related to gonadal development, were significantly expressed in ovaries of fugu (*Takifugu rubripes*) and zebrafish (*Danio rerio*) [46,47]. It facilitated sex differentiation and gonadal development in medaka and Japanese flounder [48,49,50]. Besides, prominent expression of *sox9b* was detected in gonads of early-stage *C. semilaevis*, indicating its potential involvement in gonadal differentiation [51]. *Pou5f1* (also known as *oct4*) is a key TF regulating embryonic stem cell pluripotency, primordial germ cell formation, early embryonic and gonadal germ cell development [27]. *Pou5f1* analogue in teleost plays a post-embryonic role in adult gonad and gametes development [52,53,54]. Based on the suppressive effect of *Cs-ewsr1-w* knockdown on several sex-related genes, we proposed that it might be a positive regulator in ovarian development of *C. semilaevis*. In the future, analysis including in vivo trials would be conducted for further investigation on its mechanism.

Promoters contain sequence-specific binding sites for many TFs, and the TFs could recognize and bind target sequences to guide underlying transcription and regulate transcriptional activity [55,56,57]. In *Cs-ewsr1-w* promoter -733 to -154 bp was the core region that had a great effect on transcription regulation. After site-direct mutagenesis on the TF-binding sites, the activity of TF Mafk-binding site decreased significantly, suggesting potential involvement of Mafk in *Cs-ewsr1-w* transcription. Mafk, a member of small MAFs family, was essential for mice embryonic development [58]. It regulates genes involved in several cellular processes, including ubiquitination/proteasome [59]. Numerous ubiquitin-conjugating enzyme genes showed female-biased gene expressions in early developmental stages of *C. semilaevis*, revealing the indispensable involvement of ubiquitination pathway in female differentiation [15]. Based on the above analysis, we deduced that TF Mafk might be a positive regulator in *Cs-ewsr1-w* transcription during *C. semilaevis* ovarian development. The regulation mechanism between them are worthy for further functional studies. In the future, further studies will be performed to get a clearer picture on the regulation of *Cs-ewsr1-w* by TF Mafk during female differentiation and ovarian development.

## 5. Conclusions

In this study, two *ewsr1* genes were cloned and characterized from Chinese tongue sole (*Cs-ewsr1-w* and *Cs-ewsr1-z*). The female-biased gonad expression of *Cs-ewsr1-w* was observed from 90 d to 3 y, suggesting its potential roles in ovarian development. Its knockdown significantly down-regulated the expressions of *foxl2*, *sox9b* and *pou5f1*. The activity analysis, and the prediction and verification of transcription factors for *Cs-ewsr1-w* promoter shed some lights on the transcription regulation of this gene. Our findings suggested the potential roles of *Cs-ewsr1-w* in *C. semilaevis* ovarian development, providing fundamental information for further exploration on its biological functions in teleost.

## Figures and Tables

**Figure 1 animals-12-02503-f001:**
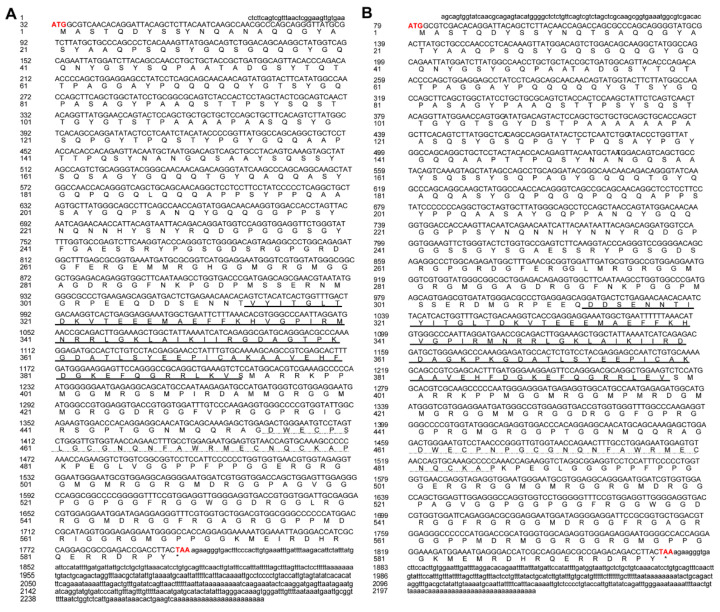
The ORF and predicted amino acid sequences of *Cs-ewsr1-w* gene (**A**) and *Cs-ewsr1-z* gene (**B**). The UTR region sequence is represented in lowercase letters. The ORF sequence is represented in uppercase letters. The start codon and stop codon are bold in red. An asterisk (*) represents the stop codon at the end of the ORF. The RNA recognition motif is represented by an underscore and the Ran binding protein zinc finger domain is represented by a dot-dash underline.

**Figure 2 animals-12-02503-f002:**
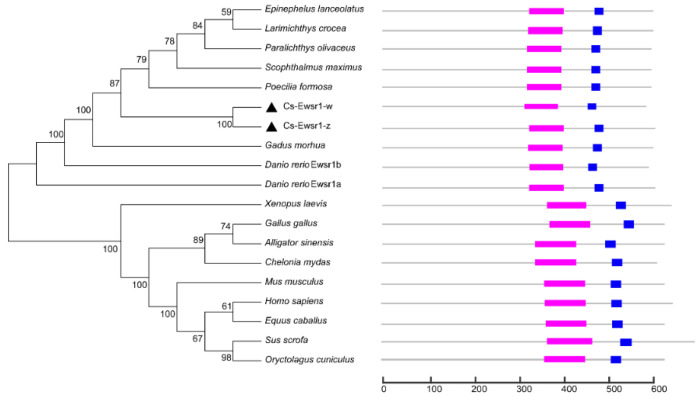
Phylogenetic analysis of Ewsr1 proteins in multiple species. “
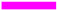
” represents RNA recognition motif, “
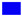
” respresents Ran binding protein zinc finger domain, “▲” represents Ewsr1 proteins in *C. semilaevis*.

**Figure 3 animals-12-02503-f003:**
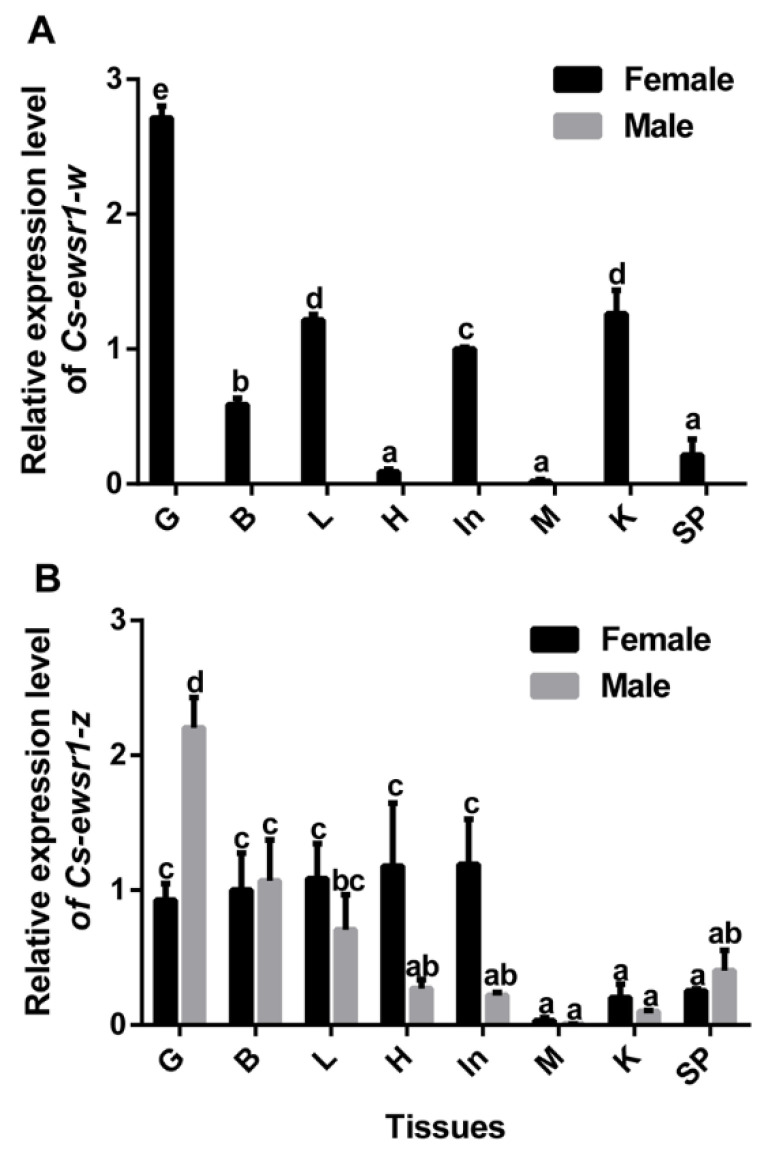
The gene expression patterns of *Cs-ewsr1-w* (**A**) and *Cs-ewsr1-z* (**B**) in various tissues of healthy C. semilaevis, including gonad (G), brain (B), liver (L), heart (H), intestines (In), muscle (M), kidney (K), and spleen (SP). Different letters represent significant differences among species (*p* < 0.05).

**Figure 4 animals-12-02503-f004:**
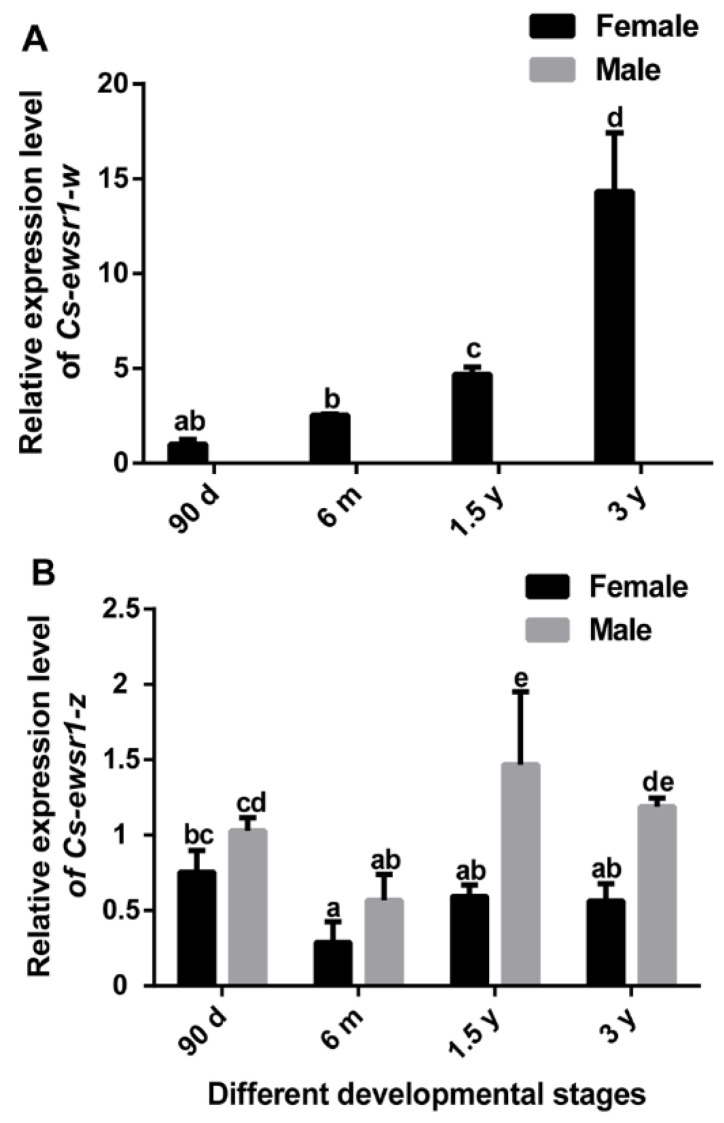
The relative expression patterns of *Cs-ewsr1-w* (**A**) and *Cs-ewsr1-z* (**B**) genes in female and male gonads from different developmental stages. Different letters represent significant differences among species (*p* < 0.05).

**Figure 5 animals-12-02503-f005:**
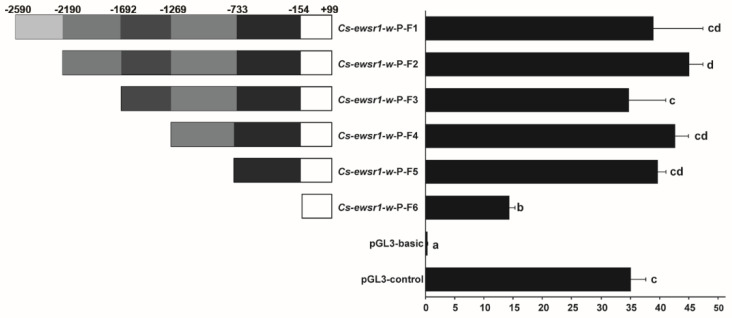
Fluorescence activity of different fragment lengths in the promoter region of *Cs-ewsr1-w* gene. The left panel exhibited the schematic map of promoter fragments with different deletion regions. The right panel showed the luciferase activities of the promoter fragments with different deletions. Different letters represent significant differences among species (*p* < 0.05).

**Figure 6 animals-12-02503-f006:**
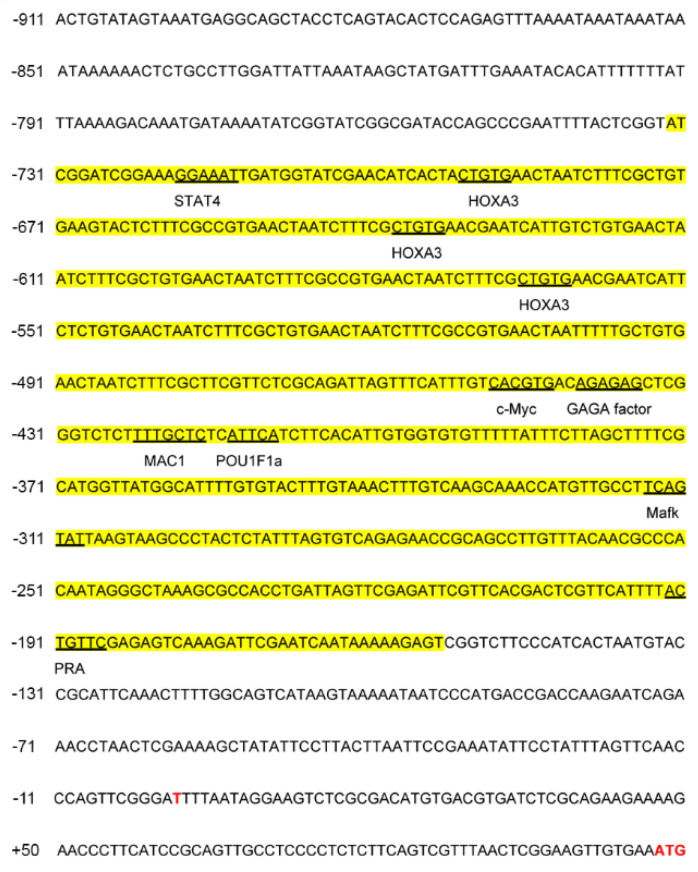
Nucleotide sequence of *Cs-ewsr1-w* promoter region (−911/+50) and the predicted transcription factor binding sites. The highly active region (-733/-154) was highlighted in yellow. Star codon “ATG” was labelled in red. Underlined boldface letters indicated the predicted transcription factors.

**Figure 7 animals-12-02503-f007:**
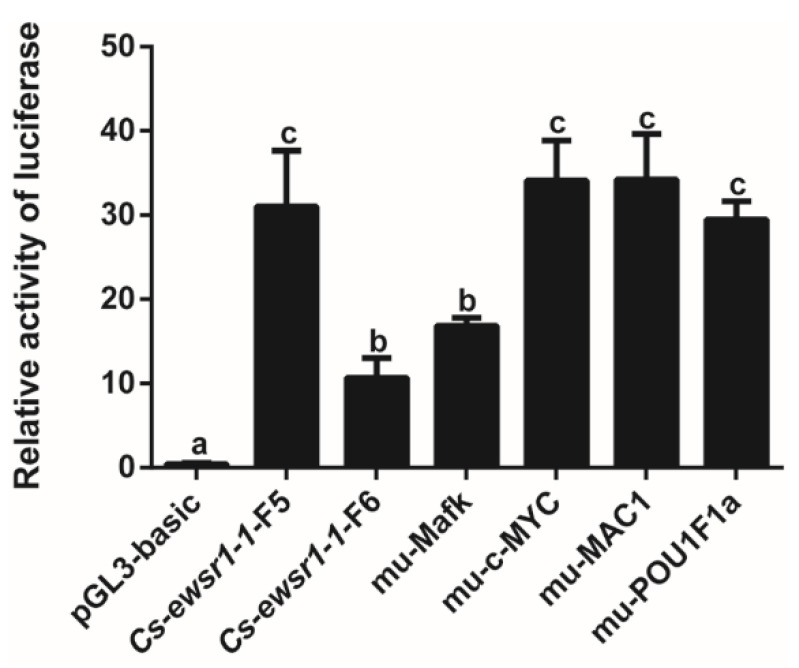
Fluorescence activity of the mutated transcription factor binding sites in *Cs-ewsr1-w* promoter compared with that of *Cs-ewsr1-w* promoter region with different deletion. Different letters represent significant differences among species (*p* < 0.05).

**Figure 8 animals-12-02503-f008:**
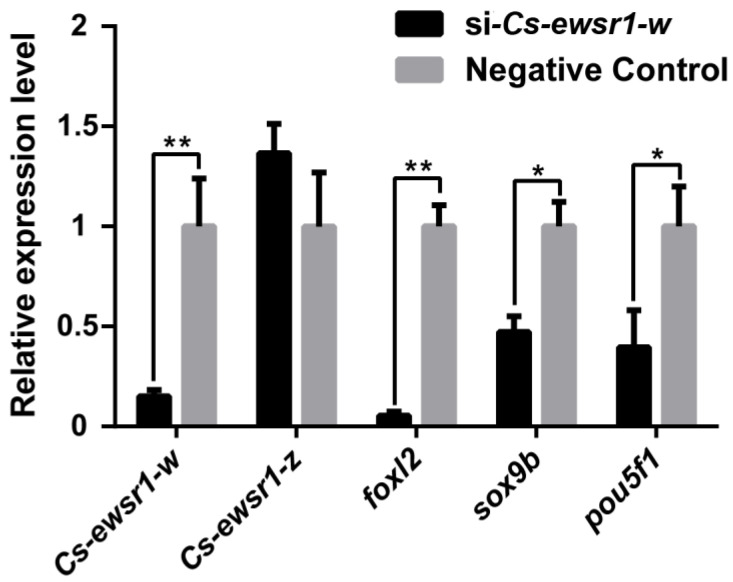
The knockdown effect of *Cs-ewsr1-w* siRNA in *C. semilaevis* ovarian cells. The relative expression variations of *Cs-ewsr1-z*, Forkhead Box L2 (*foxl2*), SRY-box transcription factor 9b (*sox9b*), POU Class 5 Homeobox 1 (*pou5f1*) were measured. An asterisk (*) indicates a significant difference between negative control group and siRNA-treated group (*p* < 0.05). Two asterisks (**) indicates an extremely significant difference (*p* < 0.01).

**Table 1 animals-12-02503-t001:** All primers used in this study.

Symbol	Information	Sequences
*Cs-ewsr1-w*-F	CDS cloning	TTCAGTCGTTTAACTCGGAAGT
*Cs-ewsr1-w*-R	CDS cloning	TCACAAGTGGGAAAGTCACCCT
*Cs-ewsr1-w*-5′-1	5′-UTR	CTGGAGCAGCAGCAGCTGGAGTACT
*Cs-ewsr1-w*-5′-2	5′-UTR	TAACCCTGCTGGGCGTTGGCTTGA
*Cs-ewsr1-w*-3′-1	3′-UTR	AATGAGAGGCAGCATGCCAATAAGA
*Cs-ewsr1-w*-3′-2	3′-UTR	GGTCGGCGGTCCTCCATTCCCCCCT
*Cs-ewsr1-w*-RT-F	qPCR	GCGGGCCCCCCATGGACC
*Cs-ewsr1-w*-RT-R	qPCR	CAAATTTCACAAGTGGGA
*Cs-ewsr1-w*-P-F1	promoter	AGATCTGCGATCTAAGTAAGCTGACGCTGGCATGTATGTT
*Cs-ewsr1-w*-P-F2	promoter	AGATCTGCGATCTAAGTAAGCTGAGGACCACAACGACCCA
*Cs-ewsr1-w*-P-F3	promoter	AGATCTGCGATCTAAGTAAGCTTGCGAACAAATCACTGCG
*Cs-ewsr1-w*-P-F4	promoter	AGATCTGCGATCTAAGTAAGCTAGGTGTATCCTAAACAGAAA
*Cs-ewsr1-w*-P-F5	promoter	AGATCTGCGATCTAAGTAAGCTATCGGATCGGAAAGGAAA
*Cs-ewsr1-w*-P-F6	promoter	AGATCTGCGATCTAAGTAAGCTCGGTCTTCCCATCACTAA
*Cs-ewsr1-w*-P-R	promoter	CAACAGTACCGGAATGCCAAGCTTTCCGAGTTAAACGACTGAAGA
mu-Mafk-F	TF binding site mutation	CAAACCATGTTGCCTCGTACGCTAAGTAAGCCCTACT
mu-Mafk-R	TF binding site mutation	CAAACCATGTTGCCTTCAGTATTAAGTAAGCCCTACT
mu-c-Myc-F	TF binding site mutation	ATTAGTTTCATTTGTAGACGAACAGAGAGCTCGGGT
mu-c-Myc-R	TF binding site mutation	ACCCGAGCTCTCTGTTCGTCTACAAATGAAACTAAT
mu-MAC1-F	TF binding site mutation	AGAGCTCGGGTCTCTACGATGATCATTCATCTTCACA
mu-MAC1-R	TF binding site mutation	TGTGAAGATGAATGATCATCGTAGAGACCCGAGCTCT
mu-POU1F1a-F	TF binding site mutation	AGTAAGCCCTACTCTCGGCTGTGTCAGAGAACCGC
mu-POU1F1a-R	TF binding site mutation	GCGGTTCTCTGACACAGCCGAGAGTAGGGCTTACT
*Cs-ewsr1-w*-siRNA	siRNA	CGUUUAACUCGGAAGUUGUGA
*sox9b*-F	qPCR	AAGAACCACACAGATCAAGACAGA
*sox9b*-R	qPCR	TAGTCATACTGTGCTCTGGTGATG
*foxl2*-F	qPCR	GAGGAAGGGCAACTACTGGA
*foxl2*-R	qPCR	CAGCGACCAGGAGTTGTTCA
*pou5f1*-F	qPCR	CCATCTGCCGCTTTGAGG
*pou5f1*-R	qPCR	CCTGGGTGTTGGGTTTGG
*Cs-ewsr1-z*-F	CDS cloning	ATGGCGTCGACACAGGATTACAGCT
*Cs-ewsr1-z*-R	CDS cloning	TTAGTAAGGTCTGTCTCGGCGCTCC
*Cs-ewsr1-z*-5′-1	5′-UTR	CAGCAGCTGGAGTACTGTCATATCC
*Cs-ewsr1-z*-5′-2	5′-UTR	CCCCTGCTGGGCGCTGGTCTGG
*Cs-ewsr1-z*-3′-1	3′-UTR	GATGAGAGGTGGCATGCCAATGAGA
*Cs-ewsr1-z*-3′-2	3′-UTR	AGGCGGAGGTCCTCCATTTCCCCCT
*Cs-ewsr1-z*-RT-F	qPCR	GGATATGACAGTACTCCAGCT
*Cs-ewsr1-z*-RT-R	qPCR	TCCTGCAGGCTGGCTATAGCTAC
sex-F	sex identification	CCTAAATGATGGATGTAGATTCTGTC
sex-R	sex identification	GATCCAGAGAAAATAAACCCAGG

**Table 2 animals-12-02503-t002:** Accession numbers of Ewsr1 proteins used in this study.

Species	Accession No.
*Larimichthys crocea*	XP_019130992.1
*Paralichthys olivaceus*	XP_019956591.1
*Poecilia formosa*	XP_007559175.1
*Gadus morhua*	XP_030215256.1
*Scophthalmus maximus*	XP_035495384.1
*Danio rerio-*Ewsr1a	XP_021334784.1
*Danio rerio-*Ewsr1b	NP_997795.1
*Epinephelus lanceolatus*	XP_033475149.1
*Alligator sinensis*	XP_025048938.1
*Homo sapiens*	XP_011528297.1
*Gallus gallus*	XP_015150339.2
*Mus musculus*	NP_001269990.1
*Chelonia mydas*	XP_037735001.1
*Sus scrofa*	XP_020927659.1
*Equus caballus*	XP_023502669.1
*Xenopus laevis*	XP_018095208.1
*Oryctolagus cuniculus*	XP_017206143.1

## Data Availability

The data presented in this study are available in this article.

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
