# Peer review of "Potential Involvement of ewsr1-w Gene in Ovarian Development of Chinese Tongue Sole, Cynoglossus semilaevis"

_animals, 2022, doi:10.3390/ani12192503_

Round 1

Reviewer 1 Report

In general, the present manuscript “Potential involvement of ewsr1-w gene in ovarian development of Chinese tongue sole, Cynoglossus semilaevis” by Cheng and colleagues, focused on studying the role of ewsr1-w gene in Chinese tongue sole, Cynoglossus semilaevis. The study is very interesting and provides insight for extensive work. Overall, the manuscript sounds good, designated well with enough data in the result. The paper is acceptable upon some concerns are addressed.

1.  In simple summary, line 18, replace ‘has’ by ‘have’.

2.  In simple summary, line 19, please add “the” before Ewsr1-w gene.

3.  In simple summary, line 22, its single gene, so use‘role’ instead of ‘roles’. 

4.  In line 35, there should be some words about Mafk.

5.  In line 55, “igf7 “should be “igfbp7”.

6.  In introduction, line 64, should begin with "The"

7.  In introduction, line 65, remove ‘the’ from ‘the other cellular processes’.

8.  In introduction, line 76-85, no need to discuss the whole conclusion of the study. These sentences need re-write.

9.  For regents or kits used in the paper, the companies should be indicated together with their city (states abbreviation is required for USA and Canada) and country in their first appearance.

10.In line 104, full name of “yph” should be given as it appears for the first time.

11.In line 169, a reference or detailed description is needed for the C. semilaevis ovarian cells.

12.In line 215-216, “its low expression in testis”, It should be “no expression in testis”.

13.In discussion, line 273-284, add more and latest literature about Ewsr proteins, Cs-Ewsr1-w and Cs-Ewsr1-z

14.In conclusion, line 333-341, please left clear to the reader about the main finding, future recommendations and novelty of the current work.

Reviewer 2 Report

 I have reviewed the manuscript "Potential involvement of ewsr1-w gene in ovarian development 2 of Chinese tongue sole, Cynoglossus semibreves" . After careful reading, I found the current form of the manuscript is not fit for publication. Please find the critical concerns and questions and justify the same before resubmitting it for publication in the journal Animals. 

Simple Summary

·         Some of the sentences are too hard to follow and have grammatical errors (lines 14-16, 18, 21 and 22)

·         Sexual growth dimorphism or sexual dimorphism?

·         Line 14-16 don’t understand what the authors are trying to convey, the sentences are talking about different aspects and is not linked to one another.

·         “while rare studies have been reported in teleost”, reframe the wording.

Abstract

·         Line 25, repetition of same word ‘multifunctional’

·         Line 33, what cell line is used?

·         Line 35, activity in ovary? If so in young or old fish?

Introduction

·         Line 45-46, not sure if there should be a number (20) needed in that sentence, there is a lot is species has this phenomenon

·         Line 51, Cyp19a gene is more important for sex determination or differentiation

·         Line 55 its igfbp7 not igf7

·         Line 55 “The female-biased gonad gene igf7” is igfbp5 is a gonadal (not gonad) gene?

·         Line 59 it's not “sex differentiation and determination” it is sex determination and differentiation (determination is the first thing happens during development)

·         Line 58-63, simplify into a single sentence

·         Line 66 ‘TET’, expand

·         Line 69 restructure, pay attention to how you are writing/ framing a sentence

·         Line 70, “In mice, the offspring of ewsr1-deficient” rewrite

·         Sterileness or sterile?

·         Line 64-75 feels like reporting of available information, no connections or no context, please put things in context and reframe it.

·          Line 78, expression profilings or profile?

·         Gonad development to gonadal development, change throughout

·         Line 82-83, poor sentence structure, can’t understand what the authors are trying to convey

·         Poor English language/grammar use throughout, read carefully and make appropriate changes throughout.

Materials and Methods

·         Line 99 Primers to primers (small p)

·         Did the authors quantify the RNA before cDNA synthesis, if so how and how much RNA is used to make cDNA?

·         Line 117, all primers work at 55°C?

·         Line 140-141, what are the default settings used,  need to give cycle information and annealing temperature details?

·         Promoter activities analysis is in-house assay or following a standard kit/published protocol

·         How did the authors run negative control (NC) groups for the siRNA gene knockdown study?

·         How did the authors confirm transfection and effecting gene knockdown at 48 hr time point?

Results

·         Line 186, function domains or functional domains?

·         Line 207 “It exhibited the significantly highest” incorrect wording

·         Line 230 was in where?

·         Is it possible that the expression of Cs-ewsr1-w and Cs-ewsr1-z are compensated to one another?

·         Figure 8, the way the graph plotted is confusing, is the siRNA is the test group or the negative control?

·         In figure 8 if the negative control is the test there should be a positive control to compare the expression of the siRNA group

Discussion

·         Line 291 reference missing

·         291-293, not clear what was authors are trying to convey

·         295-297, poor sentence structure

·         Line 291-313 the authors are talking about the reported gene functions of the genes they measured in this study, which is not helping to put their data in the context of the discussion. Recommending rewriting the paragraph

·         Line 34-31, the statement seems to be vague, possible explanation?

·         Last paragraph of the discussion demand refinement, particularly in discussing the results and putting the results in context and future directions.

Overall, there are critical issues with English writing, grammar and spelling. I tried to point out some which take the review's focus from scientific scrutiny to English language checking. Recommend (highly) the authors to read the manuscript throughout and correct the language and critical errors and concerns pointed out above before submitting it for publication in the journal. Also include explanations in the revised version (not just for the response document for the reviewer) for critical concerns asked.

Round 2

Reviewer 2 Report

The authors did an excellent job in revising the manuscript. Please make sure that all the questions raised by the reviewers are addressed in the manuscript (not just in response to the reviewer document) in the final version for publishing in the Journal Animals. 

Author Response

A: Thank you for your comments. We go over all the questions, and revised the correlative text in this version, most of which is in M& M. Hope this would improve the quality of our manuscript.